# Transition from Conventional Broiler Meat to Meat from Production Concepts with Higher Animal Welfare: Experiences from The Netherlands

**DOI:** 10.3390/ani9080483

**Published:** 2019-07-25

**Authors:** Helmut W. Saatkamp, Luuk S. M. Vissers, Peter L. M. van Horne, Ingrid C. de Jong

**Affiliations:** 1Business Economics Group, Wageningen University, Hollandseweg 1, 6706 KN Wageningen, The Netherlands; 2Wageningen Economic Research, Hollandseweg 1, 6706 KN Wageningen, The Netherlands; 3Wageningen Livestock Research, De Elst 1, 6708 WD Wageningen, The Netherlands

**Keywords:** animal welfare, broiler production, improved production concepts, the Netherlands

## Abstract

**Simple Summary:**

Animal welfare (AW) in conventional Dutch broiler production (i.e., production systems that only satisfy the legal minimum requirements) has been under critique for decades. Then, suddenly, in 2014–2015, AW was improved for the entire Dutch fresh meat market for broilers. A study was conducted regarding the main reasons and decisive factors of this transition. It was found that important factors in bringing-about this transition were: (1) The availability of a cost-efficient alternative to conventional concepts, (2) a basic willingness to change within the entire value chain (including consumers), (3) initiating and triggering actions by non-governmental organizations (NGOs), (4) decisive initiatives by retailers and (5) simultaneous introduction of the new concept and replacement of the conventional concept (i.e., depriving the consumer of a cheaper choice alternative). It was concluded that, if these decisive factors were present, considerable improvements with regard to AW could be obtained in a relatively short period of time. This information can be of use for value chains of other species (e.g., pork) and countries outside the Netherlands.

**Abstract:**

Since the 1970s, animal welfare (AW) in Dutch broiler production has been criticized by non-governmental organizations (NGOs) and the general public. Despite the development of production concepts aimed at improving AW, the conventional concept, which satisfied only the minimum legal requirements, remained by far the most dominant one in the Dutch fresh broiler meat market. Then, quite suddenly, in 2014–2015 (i.e., within less than two years), a new broiler concept with increased AW was introduced, which included a slower growing animal, more space, and an improved light regime. This alternative completely replaced the by then conventional concept. The aim of this study was to investigate the origin, causes, and driving forces of this sudden change. Popular and scientific literature, as well as interviews with key players in this transition process, were used to re-construct the chronology of events and draw the main and decisive findings. The latter include: (1) The availability of a cost-efficient alternative to conventional concepts, (2) a basic willingness to change within the entire value chain (including consumers), (3) initiating and triggering actions by NGOs, (4) decisive initiatives by retailers and (5) simultaneous introduction of the new concept and replacement of the conventional concept (i.e., depriving the consumer of a cheaper choice alternative). The result was a real transition of the Dutch fresh meat market without negative purchasing responses of the consumers. It was concluded that, although the Dutch fresh broiler meat market only included 30% of total domestic production, the existence of the abovementioned decisive factors could bring about an important change in favor of AW within a short period of time.

## 1. Introduction

Since the 1970s, the issue of animal welfare (AW) has received increasing public attention. It started with layer chicken kept under battery conditions and gradually expanded to broilers. Broadly, the public debate was dominated by two groups; on one hand were those who heavily criticized the housing and management of broilers and urged for improvement of AW and on the other were those that doubted that there was an AW problem at all. Occasionally, new initiatives to improve broiler AW were undertaken, however the impact on mainstream broiler production was very limited. Until roughly 10 years ago, these developments were focused on the development of the “right side” of the spectrum—development of new biological and organic broiler production systems. These systems claimed to have a high AW, but the production costs were quite high (for a cost-efficiency analysis, see Gocsik et al. [1]).

Approximately 10 years ago, the first initiatives were taken in the Netherlands to develop so-called middle-segment broiler production systems. The aim was to improve AW with relatively moderate technical changes and, consequently, a moderate increase in production costs. Hence, they would be in the middle between conventional and organic/biological systems [2]. A well-known example is the Volwaard broiler, which was developed in 2007 by a joint initiative between various stakeholders. This system included a more robust and therefore a slower growing animal, more space, and a covered outdoor area (“veranda”). At the same time, the Dutch organization for the protection of animals (called Dierenbescherming, DAP (Dutch Animal Protection), in Dutch) introduced the so-called Beter-Leven 1-2-3 star system to brand AW products. The Volwaard concept received 1 star, whereas, for instance, organic systems received 3 stars. Although Volwaard broilers were offered by the main retailers at a relatively small price premium compared to conventional broilers, a vast majority of consumers still preferred the latter. Hence, the success of this first nationwide middle segment was limited.

Then, starting in 2012, within less than two years, the entire Dutch broiler market for fresh meat changed, i.e., a breakthrough occurred in the Netherlands. Several large retailers, including market leaders Albert Heijn and Jumbo, took initiative regarding the broiler market. They decided to change their assortment by no longer supplying conventional broiler meat and replacing it with a new product from their own concept with (claimed) higher AW. Each retailer had its own brand name, e.g., “Nieuwe Standaard Kip” (“New Standard Chicken” in English, Jumbo) and “Hollandse Kip” (“Dutch Chicken” in English, Albert Heijn). Differences exist between these new Dutch retail standard (NDRS) concepts, but common features include a more robust, slower growing breed, more space, and an adapted light regime (these concepts are explained in more quantitative detail in Table 1). Although the NDRS did not meet the 1-star standards completely (see Vissers et al. [3]), this was still regarded as a major event in improving AW, particularly because the initiative came from the retailers themselves.

This rapid and complete change raised various questions, both within and outside the Netherlands, such as:How could such a *relatively* big change have been brought about within such a *relatively* short period of time (less than two years, after decades of inertion?What were the main conditions and driving forces that triggered this change?What lessons can be learned from these experiences valuable for other situations and regions?

Therefore, the aim of this study was to provide an answer to these questions.

## 2. Approach

First, all relevant information was collected, which included scientific literature in English and (in most cases) Dutch, together with professional literature (in Dutch), and completed with expert information. The basis was an internet-search, covering the years 2000–2016, using well-targeted keywords such as “plofkip”, “wakker dier”, and “vleeskuiken+welzijn”. This information was the basis for preparation of a draft report (Most of the underlying information used was found in sources using Dutch language only, such as newspapers, internet pages and reports. Because they are not readable for an international audience, such sources are not referred to; reference is only made to sources in English (at least with an English summary)). Subsequently, this draft report was read and judged by four key players, followed by a one hour telephone interview. The main aim was to obtain a confirmation of a correct factual representation or, if appropriate, a factual improvement of the draft report. The four key players all had been personally involved in the events regarding the transition, i.e., possessed first-hand information and insight. They represented the farmers’ organization (i.e., primary producers), retailers (i.e., two key players that were affiliated with Dutch retailers) and AW non-governmental organizations (NGO)s; hence, the most important levels of the value chain were included, i.e., farmers, retailers, and the general public. The sole focus was on facts, hence, if appropriate, personal views or interests were excluded from the report. The comments and suggestions provided by the key players were, if appropriate, included in a revision of the draft, which was sent to them for final comments and approval of correct factual representation. This resulted in the current paper.

## 3. The Broiler Value Chain in The Netherlands and Its Implications for Animal Welfare Decision Making

An important feature of Dutch broiler production is the export orientation; approximately 60% of the product is exported. Hence, the demands (or absence hereof) of these export markets should be carefully considered, particularly those with regard to AW and associated cost prices. The export rational, as well as the open-market situation, is a very important issue in all AW debates. As a consequence, approximately only 30% of the total broiler production is fresh meat for the domestic market. Hence, changing the *domestic supply* of fresh meat only would leave the largest part of *domestic production* unaffected.

The Dutch broiler value chain can be pictured as an hourglass, including three main groups of chain-actors. Many independent primary *producers* (broiler farmers) and many individual final *consumers* are linked by only a few slaughterhouses and processers, as well as a very few (approximately five) *retail* buyers. Any change regarding AW must come from an initiative, for instance, a change in attitude, of at least one of these actors *and* these actors must be able to push through their preference through the entire value chain.

Since the early 1980s, many initiatives have been undertaken to change consumers’ preferences toward demanding products with increased AW and/or to convince farmers to operate with higher AW standards. Some kind of change in the intrinsic attitude could be observed; consumers expressed increased interest and “willingness-to-buy” [4,5,6]. Also regarding producers, an expression of “willingness-to-produce” at higher AW levels could be observed [7]. Nevertheless, although this increasing latent consumer demand for higher levels of AW was waiting to be mobilized, the locked-in situation remained (locked-in here means despite a two-sided willingness, no movement occurred). Most likely, this was due to a mismatch between differences in the price that farmers wanted and the price that consumers were willing to pay; the price consequences of a big increase in AW were too big.

Moreover, even though there were initiatives within politics (e.g., new legislation, arising of the Animal Party), the locked-in situation remained until 2012–2014, the main reasons here being, in the end, AW had a relatively low priority and procedures take a long time, resulting in initiatives being stranded. Hence, and particularly within a European context, important changes in AW would be beyond legal requirements, i.e., would have to come from actors within the value chain.

Then, the third group of value chain actors started playing an active role, i.e., retailers, most prominently the market leaders. This group of actors, approximately five buyers of important retail companies, have a pivotal place in the value chain—they have the power to *provide products* and also to *not provide competing products*. In this way, they can influence supply and even create demand of one product and deny demand of another. In this respect it is important to note that in the Netherlands, approximately 90% of the market for fresh meat is in the hands of the retail market (i.e., only 10% of the supply in fresh broiler meat originates from small-scale private butchers and specialized shops). Understandably, a main interest of retail is to keep their market share and margin. However, besides this, intrinsic motivation and social responsibility are also factors that might, and sometimes do, impact decision-making.

The experiences over the last couple of decades with this hourglass-model of the broiler value chain regarding improvement on AW showed that the pivotal power which retail possesses outweighs that of the other two, i.e., consumers and producers. In other words, initiatives taken or supported by retailers have a higher likelihood of being successful than those of other stakeholders.

## 4. Requirements for an Increase in the Market-Share of Broiler Meat Concepts with Higher Animal Welfare

If retail is regarded as the main actor in the value chain able to bring about a change in AW (i.e., unlock the situation), influencing and/or triggering retail will only have a chance if such changes are in line with the monetary and non-monetary strategic aims of the retail themselves, i.e., a positive trade-off. The latter can mean that market-share and margin should remain as unaffected as possible, and/or that (relatively small) negative financial impacts are outweighed by positive effects on image and social responsibility, for example. Hence, starting with retail having a minimum intention to respond to social developments and demands, provided these are in their own interest as well, the following set of outside-retail requirements can be elaborated upon, which are essential for bringing about a change with a reasonable chance of success.

First, there should at least be a latent consumer demand or desire for improved AW. Usually, this is expressed as a willingness-to-pay for increased AW; however, this does not necessarily result in higher market shares of AW products. Various studies, such as De Jonge et al. [4,5], De Jonge et al. [8] and Mulder et al. [6] demonstrated such a willingness-to-pay amongst Dutch consumers.

Second, on the producers’ side, there also should be a latent willingness to move toward improved AW. Such a willingness was observed by Gocsik et al. [7] and Gocsik et al. [9]; farmers stated that they were not reluctant toward legal improvement of AW, provided that their net income was not negatively affected.

Third, improving AW inevitable would provoke increased production costs and hence require increased consumer prices [1]. As a consequence, within the existing market conditions, the space for improvement lies between these producer- (production costs) and consumer- (willingness-to-pay) determined bounds.

Fourth, elaborating the latter, production concepts should be available that have a high cost-effectiveness regarding AW, i.e., that bring about relatively large (initial) increases in AW against relatively low increases in production costs. The cost-efficiency analysis of Gocsik et al. [1] showed that, in this regard, particular attention should be paid to breed-related aspects (e.g., foot lesions), animal density, and light.

Finally, following the cost-efficiency viewpoint, focus should be on legal improvements in the middle-segment, i.e., somewhere between the two ends of the spectrum ranging from conventional on the one hand to extensive outdoor and organic/biological on the other.

The above-mentioned requirements provide the conditions for a change toward a more AW-friendly production. However, existence of these conditions would not necessarily start a change process on its own. To exploit these requirements in such a way that inertia is abandoned and retail “starts to act” (i.e., brings about a change), a trigger should be present to ignite the action. In this sense, NGOs can play a crucial role in sensitization of key actors as well as society at large. With regard to these NGOs, (at least in the Netherlands) two types can be observed:NGOs which are more consensus-oriented, e.g., DAP, which took initiatives on the 1-2-3-star system and the Volwaard concept;The more activist NGOs, e.g., Wakker Dier, which are more focused on direct action, but which are often regarded as too radical by other stakeholders (particularly producers).

Both can play a crucial role, each in their own way. Arguably, it can be stated that during the last decade within the Netherlands, regular consensus-oriented NGOs provided a much more broader and positive atmosphere toward AW within the whole of society (including all relevant stakeholders, primary producers, and consumers). This provided fertile soil for a change. In turn, the above-mentioned requirements and the fertile soil appeared to be the perfect conditions for the more activist NGOs (i.e., Wakker Dier) to act as a direct trigger towards the retailers. Their strategically well-targeted campaign (see below) appeared to be the perfect trigger for others, particularly retailers, to leave their passive role and take action.

## 5. Chronology of Events in The Netherlands

Above, both the main actors and pre-conditions were described briefly. Here, in chronological order, the main events in Dutch broiler production that were facilitated and/or triggered by these actors and pre-conditions and resulted in the current situation are concisely described.

**1970s** The issue of AW started to receive public attention and management of farm animals started to be criticized (with battery-kept layers being the first group receiving attention), frequent debates and individual initiatives started to pop up, and awareness grew (first by consumers, later also by producers). However, the rather locked-in situation without major changes or improvements remained.

**1980s** Development of alternatives to conventional broiler keeping, new production, and market concepts. These were primarily focused on extended outdoor and biological/organic systems and uptake by producers and consumers was very low, most likely because of the high price and low availability.

**1990s** The huge epizootic of Classical Swine Fever in pigs (1997–1998), very soon followed by epizootics of Foot-and-Mouth-Disease in cattle and pigs (2001) and Highly-Pathogenic Avian Influenza in poultry (2003), initiated a wave of public concern on animal production at large and triggered a large-scale public debate focused on alternatives and changes to current animal production.

**2001** The term “plofkip” (“exploding chicken”) was introduced by Wakker Dier, later followed by frequent, nationwide use.

**2006** Entry of the Animal Party into Dutch parliament, which caused an increased interest of all other parties in AW issues. In turn, this had a stimulating effect on AW issues in politics, public debates, and the media.

**2007** The EU directive 2007/43/EC [10], which laid down minimum rules for the protection of chickens kept for meat production, came into effect. This directive regulated the maximum density for broilers and it also provided standards for mortality rates (note that growth rate is not regulated in this directive). The main standards are presented in Table 1.

**2008** Introduction of the Volwaard concept, jointly developed by AW organizations (i.e., DAP), producers, slaughterhouses, and retailers (see Table 1 for details). This concept was the first nationwide attempt to serve the middle-segment market and included a slower growing and more robust animal, decreased animal density, and covered outdoor area (see Gocsik et al. [1]). Volwaard broilers were offered by the main retailers at a relatively small price premium compared to the conventional broilers. The success was limited, mainly because a cheaper alternative (i.e., the conventional broiler) was offered simultaneously in the shops and consumers preferred the cheaper alternative.

**2008** Introduction of the 1-2-3-star system by DAP; Volwaard given 1 star in the system.

**2011** The ongoing societal debate on AW resulted in stakeholders joining in the so-called “Agreement of Den Bosch”, an agreement between all stakeholders (ranging from producers to retailers, i.e., Jumbo) in the southern province of Noord-Brabant (the region with the highest animal density in The Netherlands) aiming at more sustainable livestock production; the essentials of this agreement were included later in official government policy on livestock production.

**2012** Implementation of a detailed campaign plan (which was finalized in 2011) by the NGO Wakker Dier. The campaign started with nationwide branding of the term “Plofkip” (“exploding chicken” in English) and the term was associated with bad AW. It very soon turned out to be a very catchy word (rapidly, the word was being used by everyone), having a high visualization power for the “disgusting” way broilers were housed under conventional conditions (disgusting in the sense of animal management, not in the sense of meat taste or consumption).

**2013–2015** Wakker Dier organized a more-or-less continuous intensive media campaign with consistent use of the word “Plofkip”, a high degree of visualization of animal management practices and consequent provision of alternatives, for both consumers and retailers to the exploding chicken (i.e., conventional broiler). The campaign used three distinct phases, i.e.,:Naming: Approach of processing and final-product companies (e.g., baby food, fast food) asking them to not use conventional broilers anymore;Shaming: In case of no response to the naming, publicly shaming these companies; this resulted in a very quick changes, i.e., a promise to change as soon as possible;Faming: In case of changes, publicly faming these companies and at the same time shaming those that still were reluctant.

One thing that is important to note is that these actions were not directed to (blaming) producers/farmers; instead, consumers still buying conventional products were mildly blamed (“you cannot do that anymore!”). Moreover, first actions were directed toward the luxury segment (e.g., baby food, airline catering, and fast food), followed by the fresh meat segment or retailers/supermarket [11].

**2013** The media campaign resulted in retail blaming farmers who were reluctant to improve AW, i.e., who were not adopting the already available Volwaard alternative to conventional broilers. A second result was that farmers, retailers, and slaughterhouses started working jointly on a less costly alternative to Volwaard, with focus on breed (i.e., growth rate and robustness), space, and light, i.e., a middle-way between the conventional broiler and Volwaard.

**2013** This resulted in the agreement between retail, primary producers, and animal welfare NGOs (i.e., DAP) on standards for the Chicken of Tomorrow (“Kip van morgen” in Dutch); the *collective* retail promised to provide “a better broiler” in 2020 (called the “Chicken of Tomorrow”). Earlier, it was stated that this was not possible because of the complexity of the transformation, availability of breeding material, and parent stocks for more robust and, hence, slower growing broilers. Wakker Dier blamed this initiative as being “no real improvement in AW of the broiler”.

**2014** The collective initiative on the “Chicken of Tomorrow” was more or less blocked by the anti-cartel organization in the Netherlands (Autority on Consumers and Markets (ACM)) because this initiative, which was *collectively* initiated and supported by all stakeholders, was judged as being a violation of the anti-monopoly regulations of the Netherlands and the EU. This judgement was regarded by the collective stakeholders as questionable and not in line with the government policy (see above: Agreement of Den Bosch). As a consequence, *individual* retailers started initiatives on their own.

**2014** The media campaign of Wakker Dier got more and more attention and the phrase “exploding chicken” (and associated pictures on TV) became known all over the country (moreover, the phrase “exploding chicken” was accepted in the official dictionary of the Netherlands (i.e., Van Dale), hence becoming part of both popular and official language). This created an atmosphere that “forced” the value chain, i.e., retail, to do something in order to avoid complaints regarding not caring about AW at all. One could call this a paradigm shift; for the first time, retail took the initiative themselves and went from staying away from AW to using AW as an asset for the by showing care for AW. Other stakeholders, i.e., producers and slaughterhouses, had to follow.

**2014 (May)** Albert Heijn (the biggest retailer in the Netherlands) launched a new concept called “Hollandse Kip”, claiming a much higher level of AW. Wakker Dier responded that this was hardly any improvement and brought this claim to the semi-official independent Commission on Advertisement Customs (Reclame Code Commissie in Dutch), which granted the claim, i.e., Albert Heijn was forced to withdraw their claim of a much higher level of AW.

**2014 (October)** Jumbo (main competing retailer of Albert Heijn), as a kind of counterattack, launched their new concept called a “Nieuwe Standard Kip”, which had a higher level of AW, close to 1-star. Although not fully satisfied, Wakker Dier called this development a breakthrough. In response, Albert Heijn improved their initial concept. It is important to note that both Albert Heijn and Jumbo *replaced* the conventional broiler with their own concept, hence they removed the “bottom conventional alternative” as a choice for Dutch consumers. At the same time, concepts with higher AW standards (i.e., 1-star) were still being offered to the consumers. For details on the NDRS, see Table 1.

**2015** Other retailers followed the initiatives of Albert Heijn and Jumbo, all with their own concepts. As a result, in almost the entire Dutch fresh meat market for broilers, the conventional broiler was replaced by animals with clearly a higher level of AW; the bottom level of AW in the market was lifted up.

## 6. Results and Current Situation

The very fast and almost complete change in the Dutch market of fresh meat for broilers that occurred in 2014–2015 resulted in a new situation, with the following important features:Regarding fresh meat, between 2014 and 2016, a transition occurred leading a new standard which one could call the New Dutch Retail Standard (NDRS), with relatively small differences between the concepts of the respective retailers.Most of the fresh-meat from broilers *sold in The Netherlands* originates from these NDRS concepts. In 2017 the market-share in the Netherlands of fresh meat was approximately 95% (see Appendix A, Table A1).The expectation is that, in the coming years, gradual further improvement will take place, i.e., further AW improvement of the current concepts.The conventional broiler (as an alternative to the higher AW-level concepts) has been banned by almost all retailers in the Netherlands, certainly in those shops where NDRS concepts are being offered to consumers.Nevertheless, the total volume of fresh broiler meat sold has not dropped (i.e., there has been no change in consumption between 2014 and 2016 and beyond) (see: Appendix A, Table A1), indicating that retail did not lose market-share (and presumably neither their margins). Since most of the underlying information used was found in sources using Dutch language only, such as newspapers, internet pages, and reports, particular care was taken in correct translation into English of key words; a native English speaker assisted in this work. However, because Dutch sources are not readable for an international audience, these are not referred to; reference is only made to sources in English or those having an English summary, at least.The individual retailers “did the job” (i.e., brought about the change), but the NGOs, particularly the more activist ones (e.g., Wakker Dier), provided important conditions.All in all, currently approximately 30% of the broilers are kept under beyond-conventional conditions (i.e., they have higher AW standards); these animals are for the fresh meat market in the Netherlands. The remaining part (i.e., less than 70% of the original total production since total kg output dropped because of a partial transition to reduced production efficiency) is still kept under conventional conditions and are for food service and export markets.Still, the overall pattern of production concepts for broilers that exists in the Netherlands ranges from conventional, via NDRS and Volwaard, to organic/biological and extreme outdoor; however, within a few years, the part of new concepts has increased from 0% to approximately 30%.In Table A1 (Appendix A), some important market features related to the transition are presented. As can be seen, domestic consumption remained the same; because in the short run total square meter production capacity remained the same, as a result of the transition, total production and export decreased. Moreover, an increase in both producer and consumer price can be observed.An important side-effect of this partial transition toward more robust animals is that the use of antimicrobials was slightly reduced between 2015 and 2016 (source: Stichting Diergeneesmiddelenautoriteit, SDa, the National Monitor on Veterinary Drug Use); although a causal relation between reduced antimicrobial use and the transition could not be established scientifically, the most obvious explanation is that the NDRS animals are more robust, hence less susceptible to disease and therefore require less antimicrobials, i.e., the NDRS concepts have contributed to this reduction.

## 7. Discussion, Conclusions, and Outlook

In this study, a specific case of the transition of an entire fresh meat market toward improved AW has been described: The complete upward shift of the bottom segment in Dutch fresh meat of broilers. This transition occurred in a very short time-span of less than two years. According to the authors, no similar examples can be found, which makes this transition rather unique.

Several distinct phases can be distinguished, each having specific features and key actors:Preparatory: A period, which lasted for several decades, in which the basic requirements, sometimes slowly, were developed, i.e., willingness of all stakeholders to (get) involve(d) in a change. Examples include a latent consumer demand for AW, the latent producers’ willingness to produce, and societal interests of retail.Getting concrete: Here, concrete reference points become visible, such as concepts (e.g., the middle segment), concrete measures (e.g., the 1-2-3-system) and/or concrete cost-efficient decision alternatives (e.g., the Volwaard concept). Consensus orientation of all stakeholders is important, since all options are, in one way or another, a compromise between AW and price, which importantly lowers the thresholds for all.Acceleration phase: Because of a catalytic action (i.e., the “Plofkip” campaign by Wakker Dier) and/or crucial stakeholders taking the initiative (i.e., retailers Jumbo and Albert Heijn announcing their initiatives with regard to NDRS), developments accelerated and the locked-in situation was unlocked. An upward pressure could occur between retailers, involving all others (i.e., producers and consumers), resulting in a rapid overall transition.

From this Dutch experience, the following general conclusions concerning the entire value chain can be drawn:A situation that has been locked-in for decades can be made fluid, provided essential requirements are met, such as latent consumer demands, willingness-to-pay for AW, and the availability of middle-market concepts that have a relatively high cost-effectiveness with regard to improvement of AW.Changes should be beyond legal and be carried out by private sector actors, since political and/or legislative actions usually take a long time; although active direct government involvement is not required or even desired, permanent background government involvement has advantages, particularly with regard to (1) keeping the dynamics of the process and (2) safeguarding level playing fields between stakeholders.Collective initiatives of the (main) chain actors entail the risk of violating the anti-cartel regulations and consequently run the risk of becoming prohibited by law.

With regard to retail, the following conclusions can be drawn:The most obvious actor to initiate change is retail for the domestic market, particularly individual retailers themselves (within the framework of their own social responsibility or just for competitiveness reasons).One important aspect to note is that, once alternatives were provided to consumers, the conventional alternative was not provided simultaneously in the same shop.

With regard to the specific role of NGOs, the following conclusions can be drawn:NGOs can play an important, if not crucial, role in initiating the change (i.e., in provoking retailers to act), for example by:Providing the right conditions for retailers to act;Framing the message using a clear sound-bite (“exploding chicken”) with corresponding visual material;Organizing well-focused media campaigns targeted to the right audience (e.g., retail (“you can do something”) and consumers (e.g., “still buying exploding chicken, you cannot do that anymore”), provided that these are not against actors as such (particularly consumers and primary producers), but are focused on those actors that are able to make changes and will be famed once they do so.Such campaigns should be aimed at creating an atmosphere that others (i.e., retail) respond to in regard to feeling the need to do something (for whatever reasons).Even campaigns that look successful can have rather limited (but promising) results; in terms of overall quantities, most of the broilers produced in the Netherlands are still produced under conventional conditions.

Out of the above, the following factors appear to be crucial:Availability of a cost-efficient alternative to conventional broilers which suits producers, consumers, and retailers; this results in a joint initiative of all stakeholders;Massive sensitization of consumers, producers, and retailers: NGOs play an important role;The ultimate initiator; one or more retailers that start to act, which forces other retailers and stakeholders to follow;Absence of alternatives (i.e., conventional broilers) in the shop, together with an affordable NDRS; retailers “cut-off” the choice range of consumers, thereby preventing substitution purchases of other products or at other retailers.

Compared to a few years ago, with regard to AW, quite a remarkable change has been achieved in the Netherlands. All parties involved look back with a mix of pride and satisfaction, as well as recognizing areas for improvement, including further development toward 1-star as a nationwide bottom level (NGOs) and gaining more control over their own affairs (including profit margin for producers). Overall, this swift transition showed that, provided basic requirements are available, a collective effort can result in an improvement in AW, a much more satisfying way of keeping broilers, an improvement in the reputation of all stakeholders, and, from the viewpoint of AW, an increase in sustainability of the broiler sector.

Nevertheless, so far this only applies to the domestic market, i.e., approximately 30% of the production. However, in important export countries, particularly Germany, similar developments have started, which offer the pursuit of further upscaling AW broiler production in the Netherlands.

## Figures and Tables

**Table 1 animals-09-00483-t001:** A description of the three main broiler production systems in the Netherlands from 2015 (source: Vissers et al. [3]).

Production System Attributes	Unit	Production System		
		Conventional	NDRS	Extensive Indoor
Broiler type		Fast-growing	Slow growing	Slow-growing
Length growth period	day	40	46	56
Weight at delivery	g	2300	2400	2300
Growth rate	g/day	60.5	49	45
Outdoor		No	No	Covered veranda ≥ 20% of total area
Stocking density	kg/m^2^	42	38	25
Natural light		No	No	Yes
Enrichment Provision of grain in feed, straw bales, or perches		No	2 g/broiler	2 g/broiler 1 bale/1000 broilers

		No	Yes	
Light intensity	lux	20	20	20
Dark period	hours/day	6	6	8
Floor type		Litter	Litter	Litter

NDRS: new Dutch retail standard.

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
