# Peer review of "Transition from Conventional Broiler Meat to Meat from Production Concepts with Higher Animal Welfare: Experiences from The Netherlands"

_animals, 2019, doi:10.3390/ani9080483_

Round 1

Reviewer 1 Report

Please check the manuscript thoroughly for slight English language errors in places.

Line 16 - what are conventional concepts?

Line 21 - likewise what are other supply chains?

The simple summary could be revised to make some of the terminology clearer

Abstract - again, conventional and new concept require explanation

Line 53 - it should read 'the' first initiatives

Line 59 - explain DAP for a non-Dutch audience

Line 66 - should read 'a' sudden break through

Line 68 - 'skipping' is not the right word, perhaps 'not supplying'

For footnote 1 it would be useful to have a comment as to whether there were any differences between the English and Dutch language texts.

Expand the methodology as currently it is not clear how the research was conducted. I.e. how was the relevant literature obtained, what searches or process was used and over what time frame. Why just English and Dutch pieces, and why not refer to Dutch articles. It would also be useful to know how data was extracted from these, and how the relevant experts were approached.

Line 85 - it would be useful to know the breakdown of expertise of those consulted. 3 types of experts are listed yet 4 are sample so it would be useful to know where the duplication is.

Line 115 - should read 'the' main reasons

Line 121 - should be 'have' not has

Line 151 - can you expand upon willingness of producers to change from the articles cited e.g. what was their motivations for doing so

Line 218 - what led to Den Bosch

Line 228 - provide information as to who was behind the media campaign as currently this is not clear

Line 273 - her should be a

Line 300 - it would be interesting to know if the market share of the types of broiler products has changed within retailers in addition to their own market share staying the same e.g. organic

Line 344 onwards - ensure consistency in captitalisation for the bullet points

It could also be beneficial to shorten this list or make it clearer. For example, could you categorise these recommendations by stakeholders involved?

The authors could also consider representing some of their findings in diagrams/tables to make them clearer for readers to see, including some of the concepts presented.

Author Response

Dear Editor and Reviewers,

Many thanks for reviewing our manuscript! Below, we provide you with our response to the comments both reviewers made. We hope we have addressed these comments satisfactorily.

On behalf of the co-authors,

HW Saatkamp (corresponding author).

Reviewer 1

R1: Line 16 - what are conventional concepts?

>Response: we’ve added some explanation in the text: ... conventional Dutch broiler production (i.e. production systems that only satisfy the legal minimum requirements)...

R1: Line 21 - likewise what are other supply chains?

>Response: we’ve added some explanation in the text: ... can be of use for value chains of other species (e.g. porc) ...

R1: The simple summary could be revised to make some of the terminology clearer

>Response: we revised the text a little, see responses on previous two comments.

R1: Abstract - again, conventional and new concept require explanation

>Response: we did the same as with the simple summary, and added a few words for more explanation.

R1: Line 53 - it should read 'the' first initiatives

>Response: we’ve changed this accordingly.

R1: Line 59 - explain DAP for a non-Dutch audience

>Response: we’ve did this accordingly.

R1: Line 66 - should read 'a' sudden break through

>Response: we’ve changed this accordingly.

R1:Line 68 - 'skipping' is not the right word, perhaps 'not supplying'

>Response: we’ve changed this accordingly.

R1: For footnote 1 it would be useful to have a comment as to whether there were any differences between the English and Dutch language texts.

>Response: we’ve revised the footnote accordingly.

R1: Expand the methodology as currently it is not clear how the research was conducted. I.e. how was the relevant literature obtained, what searches or process was used and over what time frame. Why just English and Dutch pieces, and why not refer to Dutch articles. It would also be useful to know how data was extracted from these, and how the relevant experts were approached.

>Response: we agree to this comment. We’ve included a separate ‘ Approach’  section, with more detailed explanation on the approach we’ve followed, aiming to address the reviewers comments and suggestions.

R1: Line 85 - it would be useful to know the breakdown of expertise of those consulted. 3 types of experts are listed yet 4 are sample so it would be useful to know where the duplication is.

>Response: this has been done accordingly.

R1: Line 115 - should read 'the' main reasons

>Response: we’ve changed this accordingly.

R1: Line 121 - should be 'have' not has

>Response: we’ve changed this accordingly.

R1: Line 151 - can you expand upon willingness of producers to change from the articles cited e.g. what was their motivations for doing so

>Response: we’ve added some additional text: ... farmers stated that they were not reluctant towards beyond legal improvement of AW, provided that their net-income was not affected negatively.

R1: Line 218 - what led to Den Bosch

>Response: we’ve added some explanatory text: ... The ongoing societal debate on AW resulted in that stakeholders joined in the so-called ‘Agreement of Den Bosch’: an agreement between all stakeholders ...

R1: Line 228 - provide information as to who was behind the media campaign as currently this is not clear

>Response: to make this clear, we’ve changed the beginning of the sentence as follows: ... Wakker Dier organised a more or less continuous intensive media campaign ...

R1: Line 273 - her should be a

>Response: we’ve changed this accordingly.

R1: Line 300 - it would be interesting to know if the market share of the types of broiler products has changed within retailers in addition to their own market share staying the same e.g. organic

>Response: we agree to this point of view, however this is very sensitive information which retailers will never provide. That is why we included a footnote with some additional statements: ... Information on market shares is confidential, hence no information on possible shifts before and after the transition between retailers and/or broiler products are available. Nevertheless, there are no indications of such events either.

R1: Line 344 onwards - ensure consistency in captitalisation for the bullet points

>Response: we did this accordingly

R1: It could also be beneficial to shorten this list or make it clearer. For example, could you categorise these recommendations by stakeholders involved?

>Response: we tried to do this by categorizing in the text ‘general’, ‘retail’ and ‘NGO’ focused conclusions.

R1: The authors could also consider representing some of their findings in diagrams/tables to make them clearer for readers to see, including some of the concepts presented.

>Response: we agree, that is why we included another Table (Line 231, revised version) which provides a systematic and quantitative overview of the three concepts discussed in the paper: the conventional, the NDRS and the Extensive Indoor (1-star) concepts respectively.

Reviewer 2

R2: This paper describes the evolution of the broiler market, aimed at increasing AW, in the Netherlands. It provides interesting insights on the factors that have led to this result.

In my opinion, however, some information should be provided to make the paper more understandable by the international scientific community.

In fact, the knowledge of many productive aspects is given for granted, but they do not necessarily represent the reality of all countries.

>Response: see Response to R1: we have included a Table with main features of the concepts.

R2: It is necessary to provide information about the characteristics of “conventional” rearing. To my knowledge the Netherlands produced "conventional" high-density broilers (up to 42 kg / m2, maximum allowed by law). Many other countries rear at density close to 33 kg / m2 ("standard" density of the legislation)

Furthermore, this information would help to understand, as stated by the authors, "the fast and almost complete change in the Dutch market of fresh meat for broilers"

>Response: see previous comment.

specific comments

R2: Line 59 "DAP" : explain acronym

>Response: we’ve done this accordingly

R2: Line 59 "the so-called 1-2-3 star system" : is this the Beter Leven? This is the name by which it is best known in other countries

>Response: the answer is ‘yes’, we’ve added some explanatory text.

R2: lines 71-74 more details should be provided about the NDRS system (e.g. average value of space and light regime) ; more explanations should be provided on why the NDRS system does not completely meet the 1 star-standard (reference of  Visser et al. 2019  is a submitted paper!)

>Response: see previous comment, we’ve added an additional Table explaining these issues.

R2: Line 84-86 from the acknowledgment of the paper it can be deduced that the 4 key players are 2 from retail, 1 from farmer organization and 1 from animal welfare organization. It is better to explain it also in the text

>Response: we’ve added an additional section on ‘ Approach’, in which this issues is explained in more detail.

R2: Line 92-93:  "Hence trade …… point for consideration" ;  this sentence seems a bit strong; or, at least, requires a specific reference.

>Response: we’ve revised this sentence as follows (Line 109-110 in the revised version): ... Hence, the demands (or absence hereof) of these export markets should be carefully considered, particularly those with regard to AW and associated cost prices.

-      o – o – o – o -

Reviewer 2 Report

This paper describes the evolution of the broiler market, aimed at increasing AW, in the Netherlands. It provides interesting insights on the factors that have led to this result.

In my opinion, however, some information should be provided to make the paper more understandable by the international scientific community.

In fact, the knowledge of many productive aspects is given for granted, but they do not necessarily represent the reality of all countries.

It is necessary to provide information about the characteristics of “conventional” rearing. To my knowledge the Netherlands produced "conventional" high-density broilers (up to 42 kg / m2, maximum allowed by law). Many other countries rear at density close to 33 kg / m2 ("standard" density of the legislation)

Furthermore, this information would help to understand, as stated by the authors, "the fast and almost complete change in the Dutch market of fresh meat for broilers"

specific comments

Line 59 "DAP" : explain acronym

Line 59 "the so-called 1-2-3 star system" : is this the Beter Leven? This is the name by which it is best known in other countries

lines 71-74 more details should be provided about the NDRS system (e.g. average value of space and light regime) ; more explanations should be provided on why the NDRS system does not completely meet the 1 star-standard (reference of  Visser et al. 2019  is a submitted paper!)

Line 84-86 from the acknowledgment of the paper it can be deduced that the 4 key players are 2 from retail, 1 from farmer organization and 1 from animal welfare organization. It is better to explain it also in the text

Line 92-93:  "Hence trade …… point for consideration" ;  this sentence seems a bit strong; or, at least, requires a specific reference.

Author Response

(The authors gave the same response as above.)
